# Experience as a Determinant of Declarative and Procedural Knowledge in School Football

**DOI:** 10.3390/ijerph17031063

**Published:** 2020-02-07

**Authors:** Juan M. García-Ceberino, María G. Gamero, Sebastián Feu, Sergio J. Ibáñez

**Affiliations:** 1Optimization of Training and Sports Performance Research Group (GOERD), University of Extremadura, 10003 Cáceres, Spain; mgamerob@alumnos.unex.es (M.G.G.); sfeu@unex.es (S.F.); sibanez@unex.es (S.J.I.); 2Faculty of Education, University of Extremadura, 06006 Badajoz, Spain; 3Faculty of Sports Science, University of Extremadura, 10003 Cáceres, Spain

**Keywords:** tactical games approach, direct instruction, gender, experience, physical education

## Abstract

The study of declarative and procedural knowledge makes it possible to ascertain what cognitive processes are like during motor learning. This study aimed to compare, according to the methodology, gender and experience (football practise), and the levels of declarative and procedural knowledge after the implementation of two intervention programmes on school football including one based on the tactics learning and the other on the technique learning. A total of 41 students in the 5th year of primary education from a state school from Spain, distributed in two class groups, participated in the study. Each class group participated in a different intervention programme. The sample of subjects was equal (tactical programme (*n* = 20) and technical programme (*n* = 21)). A panel of 13 experts validated both programmes. Levels of knowledge were measured using the Tactical Knowledge Assessment test in football. A descriptive analysis was performed to characterise the sample. Moreover, a t-test for independent samples, a t-test for related samples, and a *2 × 2 ANOVA* (analysis of variance) were performed to compare the levels of knowledge between the pre-test and the post-test, according to the methodology, gender, and experience of the students. Results indicate that both intervention programmes induced higher levels of declarative and procedural knowledge in the post-test. Similarly, there were no significant differences with regard to the applied methodology. This fact is due to the heterogeneous character of the class groups with gender and experience showing effects on the levels of knowledge. The boys possessed greater experience and a higher level of knowledge compared to the girls.

## 1. Introduction

The teaching of invasion sports in primary education should be oriented toward educational objectives. Within this wide group of sports, football is the collaboration-opposition sport, which is most commonly taught in the school context in Spain [1]. Thus, the importance that football demands socially and at school leads to analysing the teaching-learning processes involved [2].

The physical education teaching approach must be in learning and, therefore, how sports are taught in physical education is as important as its content [3]. The teaching of invasion sports has traditionally focused on the technical element, leaving the tactical element to one side [4]. This implies the existence of deficits in the teaching-learning process regarding cognitive structures [5]. Following this line of thought, several studies have analysed the Student-Centered Approaches (SCAs) or tactics and the Teacher-Centered Approaches (TCAs) or technique, with the idea of contrasting which teaching methodologies induce higher levels of technical-tactical knowledge with significant differences being observed in the variables related to understanding the game like declarative knowledge or decision-making in favour of the SCAs [6,7]. Within the SCAs, the Tactical Games Approach (TGA) method stands out, and, within the TCAs, the Direct Instruction (DI) method is the most common [8].

The TGA method gives students the opportunity to learn through the real game. The teacher generates the learning through questions and interrogative feedback while encouraging decision-making. On the other hand, in the DI method, the students participate in technical components or drills. The teacher gives direct information to the students including prescriptive feedback, and they execute according to the guidelines given [9,10]. The TGA method is more successful than the DI method because it bases its learning on small-sided games. The students play with modified rules, fewer students, in tight spaces, etc., which improves tactical understanding when making decisions [11]. The DI method causes little learning progression because the students participate in isolated and descontextualized tasks of the real game [12]. For all these reasons, the TGA method is more appropriate for teaching invasion sports in physical education classes.

In the school context, several studies compare methodologies in different aspects like learning [7,13], and external and internal load [14], after the implementation of two different intervention programmes for basketball, with one based on a traditional methodology (Direct Instruction in Basketball) and another on an alternative methodology (Tactical Game in Basketball). There is also another study on teaching methodologies after the implementation of intervention programmes for football [15].

MacPhail et al. [11] highlight the importance of teaching technical-tactical aspects in invasion sports from the SCAs, based on integrating and total teaching, rather than TCAs, centered on teaching technical skills. Perceptive and cognitive factors take on greater importance in SCAs given their influence on decision-making and the transfer of learning [16]. Light et al. [17] point out that SCAs improve the ability to game, increase students’ motivation, and provide positive affective learning experiences. Similarly, the teachers are the ones in charge of designing play tasks, which help the learners to develop knowledge and give them the opportunity to acquire interrelated technical-tactical aspects [18].

The students make decisions during sports practice thanks to the structures of prior knowledge that they possess and their ability to process new information [19,20]. In the same way, the students possess a great capacity to acquire knowledge at an early age [18]. Moreno et al. [21] indicate two types of knowledge: declarative and procedural. On the one hand, declarative knowledge refers to theoretical information that includes both technical-tactical aspects and rules. On the other hand, procedural knowledge refers to (tactical) decisions made in concrete game situations. Chatzipanteli et al. [9] indicate that students must have adequate declarative knowledge to improve procedural knowledge. Declarative knowledge must be conducted prior to problem solving and decision-making.

The level of the students’ knowledge and their ability to make decisions are fundamental factors for effective sports practice [22]. Tactical knowledge will favour the development of decision-making skills during sports practice [23]. Similarly, an adequate base of prior knowledge together with different factors such as experience or formal instruction will help the students make game decisions in a more efficient manner [24]. Williams et al. [25] and García et al. [26] indicate that declarative and procedural knowledge are acquired through sports practice, differentiating the expert students from the novices in the structured memory, which they acquire with the practice of sport and the specific knowledge that they possess. For this reason, different studies conclude that the levels of knowledge and the experience of the students are intimately linked with students with more experience presenting higher levels of knowledge [27,28,29].

Regarding gender, both in the context of sports training and of the school, boys are associated with sports that involve physical contact, strength, and aggressiveness: football, basketball, handball, etc. On the other hand, girls are associated with sports related to rhythm and beauty such as gymnastics, figure skating, dance, etc. [30,31]. In spite of studying gender with respect to different variables, especially psychological ones [32,33], the research on differences in sports learning according to gender are scarce [34].

In general terms, experience (sport practice) and gender are two mediating variables that can explain the differences in the levels of declarative and procedural knowledge recorded in physical education classes regardless of the teaching method applied by the teacher. In the Serra-Olivares [34] study, boys recorded higher levels of declarative and procedural knowledge than girls. In addition, the students with the most experience also recorded better results. This same author indicates that more studies are needed to analyze the relationships between experience, knowledge, skill, and gender in school sports.

The scientific literature includes research on the declarative and procedural knowledge of young students, after being taught football, in the school context [34,35,36] and in the context of sports training [18,25,37]. Research on football tends to study elite players [38]. In addition, many interviews, videos, questionnaires, and written tests are used as tools to evaluate the knowledge of learners in football [39].

Studying declarative and procedural knowledge makes it possible to ascertain students’ cognitive processes [2]. This knowledge is of great importance in the teaching-learning process of an invasion sport such as football [39]. The literature does not include studies, which, as well as analysing declarative and procedural knowledge, take into account the gender and level of experience of the students. The teaching of football in the Spanish educational system is common since it is necessary to study the influence exerted by the experience of the out-of-school activity in the school context. Thus, the purpose of this study was to compare, according to the teaching method, gender and experience (football practise), the declarative and procedural knowledge acquired by two groups of students from the 5th year of a state school in Spain after the implementation of two intervention programmes on school football. Each intervention programme was based on a different method, TGA and DI, and each programme was applied in a different class group.

## 2. Materials and Methods

### 2.1. Design

A quasi-experimental longitudinal design was used with a pre-test and a post-test [40] to determine if there were differences in the level of declarative and procedural knowledge after the implementation of two intervention programmes on school football.

### 2.2. Sample

A total of 41 students, aged 10 or 11 years (Age: 10.63 ± 0.488 years), from the 5th year of primary education from a state school in the west-central region of Spain, participated in the study. They were divided into two class groups depending on the intervention programme applied. The distribution by gender in each class in the Spanish state system is mixed and heterogeneous, and it is organized by the school’s academic authorities. The implementation of the programmes to the groups was random. The students from the 5th year group A participated in the Tactical Games Approach Soccer (TGAS), and the students from the 5th year group B in the Direct Instruction Soccer (DIS) intervention programme. To be able to take part in the study, it was necessary for the parents or legal guardians to sign their informed consent. Moreover, this study was included in the school’s curricular project.

The students had no previous contact with the invasion sport of football in their physical education classes, although 15% of the students who participated in the TGAS intervention programme and 42.9% of those who participated in the DIS intervention programme practised football as an out-of-school activity (or as sports training) 4 hours per week, which was divided into 3 hours training and a one-hour official match. All these students were boys. None of the girls from either group practised football as an out-of-school activity. Moreover, 40% of the students who participated in the TGAS programme and 71.43% of the students who participated in the DIS programme were boys.

This heterogeneous character of the class groups regarding level of practise and gender was conditioned by the random distribution of the students, according to the Spanish educational system, which does not segregate students according to gender. The groups were not modified to maintain the ecological validity of the study.

The characteristics of the students participating in the study are shown in Table 1.

### 2.3. Variables

Three independent variables were determined: i) the TGAS (based on the TGA method) and the DIS (based on the DI method) intervention programmes were designed in a similar manner, but were based on different methodologies. Both intervention programmes were similar (*p* > 0.05) with a high degree of association in different variables, in terms of: the number of tasks, the number of sessions, the phases of play, the specific contents, and the didactic objectives. Differences were identified in the rest of the studied variables due to the particularities of each teaching methodology, i.e., game situation, teaching means, level of opposition, degree of opposition, density of the task, competitive load, and cognitive implication (*p* < 0.05) [41]. The TGAS and DIS programmes were validated by a panel of 13 judicial experts (Aiken’s V ≥ 0.69, α = 0.97). To be considered as an expert judge, they had to meet a series of criteria: having a Doctoral degree, being a Higher Education faculty member in the area of sport pedagogy and/or invasion sports, possessing the highest federative certification level (level III) in invasion sports, having 10 years of experience or more as an invasion sports coach, or having authored publications on sport pedagogy and/or teaching methods [42], ii) the gender of the students, and iii) the practice of football as an out-of-school activity, that is the experience of the students in football.

The TGAS and DIS programmes present optimal levels of validity and reliability. Thus, both are considered valid and reliable for the teaching of football in the school context as well as for analysing the level of learning attained by the students after their implementation [42].

The dependent variable for the study was the levels of declarative and procedural knowledge [21] acquired by the students after the implementation of the intervention programmes. Both types of knowledge are related to the attack phase in football.

### 2.4. Instruments

The declarative and procedural knowledge of the studied students was analysed using the Tactical Knowledge Assessment test in Football (TCTOF by its Spanish acronym). This instrument is a written multiple-choice test divided into two parts. The first part is in written form and evaluates declarative knowledge (“to know what, and why to do it”) and the second part written with representative figures of a game situation to evaluate procedural knowledge (“to know how and when to do it”) [39]. The TCTOF is linked to the action principles of invasion sports [43].

Table 2 shows the indicators evaluated by the TCTOF for declarative and procedural knowledge [39].

Using consensual agreement [44], it was decided to eliminate nine of the 36 items aimed at evaluating declarative knowledge and one of the 16 items aimed at evaluating procedural knowledge since they presented greater complexity with regard to the educational stage of the participating students.

Lastly, the data collected using the TCTOF were exported to the SPSS 21.0 statistical program (IBM Corp. Released 2012. IBM SPSS Statistics for Windows, Version 21., IBM Corp, Armonk, NY, USA) for their later descriptive and inferential analysis.

### 2.5. Procedure

First, despite the fact that the study did not require invasive measures to obtain the data, the University Bioethics Committee was asked for its approval (Ref. 09/2018). Then authorisation was requested from the school and the physical education teachers. Once authorisation was obtained, the parents or legal guardians of the students were asked for their informed consent by signing a document that was in accordance with the ethical guidelines of the Declaration of Helsinki and Organic Law 15/1999 of 13th December on the Protection of Personal Information (LOPD) (BOE 14 December 1999). Similarly, the study was approved by the school council within the school curriculum.

After obtaining the necessary authorisations, an initial evaluation was conducted, including the pre-test, in which the students completed the TCTOF. This evaluation also included for the study purposes the following information: i) the school year (5A or 5B), ii) age in years, iii) years of football practice (1, 2, 3, 4, or more than 5), and iv) football practice in the out-of-school context (Yes or No).

After that, the TGAS and DIS programmes were implemented, with one to each class group for 11 sessions, including sessions of the 3 × 3 games of the pre-test and post-test. The teaching progressed based on the difficulty of the tasks. The teacher’s communications were also adapted to the type of feedback characteristic for each methodology [41].

Lastly, after the application of the intervention programmes, there was a final evaluation, which is the post-test, when the students again completed the TCTOF.

The pre-test and post-test assessments lasted 55 minutes each, and each time, the principal investigator explained the TCTOF, so that it was completely clear how it had to be filled out.

Figure 1, which was designed by García-Ceberino et al. [15], summarizes the structure of the TGAS and DIS intervention programmes.

### 2.6. Statistical Analyses

Firstly, the tests of assumption of criteria were conducted to identify the characteristics of the study data [45]. The Shapiro-Wilks and Levene tests showed that the study variables fulfilled the assumption of normality, so that mathematical parametric models could be used to test the hypothesis.

A descriptive analysis was then performed to ascertain the participants’ characteristics according to gender and experience in football. Then, a t-test for independent samples was performed to compare the level of declarative and procedural knowledge between both class groups (pre-test). Similarly, a t-test for independent samples was conducted to contrast the level of declarative and procedural knowledge resulting from the implementation of both intervention programmes (post-test) [45].

A t-test for related samples was also performed to contrast the level of declarative and procedural knowledge acquired by the students in each class group after the implementation of the intervention programmes (post-test) with respect to their initial level (pre-test) [45].

Lastly, to determine the effect that gender and experience in football had in the study in both methodologies on the level of declarative and procedural knowledge acquired by the students after the implementation of the intervention programmes (post-test), a *2 × 2 ANOVA* for repeated measures was performed [45], using gender and experience as co-variables.

Figure 2 presents the statistical tests used to compare the declarative and procedural knowledge in the pre-test and post-test, according to the methodology, gender, and football experience.

The effect size of the statistical analyses was determined using Cohen’s d (d_cohen_) and partial eta squared (η2) [46,47].

## 3. Results

The descriptive results for each intervention programme according to gender and experience in football are presented in Table 3. The boys who participated in each programme presented higher levels of declarative and procedural knowledge in the pre-test and post-test compared to the girls. Similarly, the students from both groups who practised football as an out-of-school activity presented higher levels of declarative and procedural knowledge in the pre-test and post-test.

Table 4 compares the level of declarative and procedural knowledge between both class groups in the pre-test and post-test. It shows that the students who participated in the DIS programme presented higher levels of initial declarative and procedural knowledge (pre-test) than the students who participated in the TGAS programme. However, these differences were not statistically significant (*p* > 0.05) with an intermediate effect size (0.500–0.799) in the level of declarative knowledge and small (0.200–0.499) in the level of procedural knowledge. After the implementation of the intervention programmes (post-test), the students who participated in the DIS programme continued to show higher levels of declarative and procedural knowledge than the students who participated in the TGAS programme. However, the differences were not statistically significant (*p* > 0.05) and showed a small effect size (0.200–0.499).

The level of declarative and procedural knowledge acquired by the students in each class group, after the implementation of the two intervention programmes (post-test), is shown in Table 5. It can be seen that both class groups present higher levels of declarative and procedural knowledge with regard to their initial level (pre-test). There were significant differences (*p* < 0.05) with a small effect size (0.200–0.499) in the level of declarative and procedural knowledge between the pre-test and post-test of the class group that participated in the DIS programme. However, there were no significant differences (*p* > 0.05) and with a small effect size (0.200–0.499) in the level of declarative and procedural knowledge between the pre-test and the post-test of the class group that participated in the TGAS programme.

Table 6 presents the effect that gender and experience in football have in the two methodologies studied on the level of declarative and procedural knowledge acquired by the students. Regarding gender, it can be seen that there were no significant differences (*p* > 0.05) and, with a small effect size (0.010–0.059), in declarative knowledge and no effect (0.000–0.009) in procedural knowledge, according to the methodology used. However, there were significant differences (*p* < 0.05) with a large effect size (≥0.140) in both types of knowledge, according to the gender, so that this variable affects the levels of declarative and procedural knowledge acquired by both class groups. Similarly, with respect to the students’ experience in football, we can see that there were no significant differences either (*p* > 0.05) and, with a small effect size (0.010–0.059) in declarative knowledge and no effect (0.000–0.009) in procedural knowledge, according to the methodology used. However, there were significant differences (*p* < 0.05) and, with an intermediate effect (0.060–0.139), in declarative and a large effect (≥0.140) in procedural knowledge in the experience variable, so that practising football in an out-of-school context also affects the levels of declarative and procedural knowledge acquired by both class groups.

## 4. Discussion

Currently, declarative and procedural knowledge are tools used to assess cognitive aspects in the evaluation of invasion sports like football [2]. This research has contributed to the study of levels of declarative and procedural knowledge resulting from the implementation of two intervention programmes, based on two different teaching methodologies, in two groups of students in primary education of a state school in Spain. The results indicate that the TGAS and DIS programmes induced improvements in the levels of declarative and procedural knowledge in both class groups. Moreover, the students who did not have knowledge of football improved and, those who did have knowledge, improved even more. The students’ gender and experience (football practice) affected the levels of declarative and procedural knowledge in the pre-test and post-test. The students from both class groups had more difficulties with regard to procedural knowledge (behaviours or strategies to be used in different play situations).

Different studies carried out on invasion sports like football [6] or basketball [7] state that the SCAs or tactics induce higher levels of knowledge. In this research, the students who participated in the DIS programme, based on technique, showed significant differences in declarative and procedural knowledge between the pre-test and the post-test, while the students who participated in the TGA programme, based on tactics, did not present significant differences in declarative and procedural knowledge between the pre-test and post-test. This may be due to the effect of the students’ gender and experience in the teaching of the invasion sport of football. In this respect, 40% of the students who participated in the TGAS programme were boys and 15% practised football in an out-of-school context while 71.43% of the students who participated in the DIS programme were boys and 42.9% practised football in the out-of-school context. None of the girls had experience in football. The boys’ percentage and experience in football was very low in the TGAS programme. Therefore, this programme did not present significant differences. Several studies have identified the fact that boys are more interested in practising football than girls both in the school playtime [48] and in an out-of-school context [49]. In this study, the students who had no knowledge of football improved equally independently of the method and, the students who did have knowledge, improved even more with the TGAS programme. The characteristics of the students who participated in the study show the heterogeneity of the groups, which may have affected the results. Different studies indicate that the multiplicity of behaviours and/or solutions that invasion sports offer within a class group with very dissimilar capacities and knowledge make the teaching task very difficult [50,51]. Spatial and reglementary adaptations are needed to facilitate the teaching and evaluation of football in the school context [52].

Other similar studies that compared the effects of implementing two intervention programmes for school basketball using different methodologies based on TGA (Tactical Game is Basketball) and DI (Direct Instruction in Basketball) found improvements with both intervention programmes, but the students who participated in the TGA programme reached a higher degree of learning [7,13]. Equally, in the analysis of the physical demands involved with the implementation of these programmes and their relation with the learning acquired by the students, it was found that the TGA methodology produced better results in the variables of an external and an internal load than the DI methodology, which permits a greater development of physical fitness in the students and better performance indicators in the game [14]. García-Ceberino et al. [15] also indicated that the TGA method favors the physical fitness of students. These results led to the recommendation that physical education teachers use a TGA method for teaching invasion sports in physical education classes [53].

Students have different levels of knowledge as a function of their experience in sports practice [27,29]. Persky et al. [54] state that the main element that differentiates novice students from the experts is decision-making. The results obtained in this study indicate that the students with more experience reached a higher level of declarative and procedural knowledge. However, the students who participated in the TGAS intervention programme learned more. According to Serra-Olivares et al. [23], the level of specific knowledge of football and the experience of the students are intimately related. The results of this investigation coincide with several studies where the structures of technical-tactical knowledge and decision-making in football were analysed in the school context including in secondary education [36] and in primary education [34] as well as in the out-of-school context [18,55]. Experience determines the learning attained by the students since knowledge is acquired by practising the sport. The boys participating in the study tend to practise out-of-school football more often than the girls, which means that the former reached higher levels of knowledge in the pre-test and post-test.

The level of declarative knowledge acquired by both groups was higher than the level of procedural knowledge. This could be due to the fact that the teaching they have received in previous years has been more focused on technical aspects, without taking into account the strategies to be implemented during the game (tactical decisions) and the difficulties for interpreting the most suitable tactical decision in each play situation [37]. This study highlights the importance of making the right decision depending on the situation that is presented and, on this basis, choosing the technical-tactical element to be implemented. Similarly, the processes of learning and decision-making are influenced by the egocentrism that characterises students of these ages, which leaves the group aspects of the game [43]. The problem of the students to interpret the most suitable tactical decisions in every game situation that is considered can be due to the tendency of teachers to use TCAs for teaching invasion sports in physical education classes [56], which suppresses creativity and decision-making.

Lastly, the gender of the students also affected the level of declarative and procedural knowledge. Boys present higher levels of knowledge than girls. Serra-Olivares [34] identified that the gender variable was determinant in the levels of declarative and procedural knowledge of students of primary education. The boys showed significantly higher levels of knowledge than the girls. This is because they tend to practise football more often. This could be because boys tend to practise sports that involve physical contact, strength, and aggressiveness such as football, basketball, handball, etc. whereas girls tend to practise sports involving rhythm and beauty such as gymnastics, figure skating, dance, etc. [30,31]. Slingerland et al. [57] indicate that playing invasion games in school (football, handball, basketball, etc.) could be a strategy for the physical education teachers to foment the perceived competence of the girls and their physical activity in this type of game.

Thus, the findings of this study could suggest that the differences in the levels of declarative and procedural knowledge of the students participating in the study are due to the effect of their gender and experience in football, which highlights the heterogeneous distribution of groups in physical education classes. The random distribution of the students, according to the Spanish educational system, conditions the heterogeneous distribution of the groups regarding the level of practice and gender. Further research is needed to delve more deeply into the study of cognitive structures of players at the sports initiation stage [27]. Following this line of thought, there are few studies that analyse declarative and procedural knowledge in the school context and, in particular, in primary education [34,58].

The students were not grouped in homogeneous groups due to the legal organization of the class groups. However, this fact allowed us to learn about the influence of gender and previous experiences in the results. Likewise, new research studies including more participants are needed to improve and provide more information.

The Spanish educational system recommends using the technical and tactical contents common to invasion sports. The transfer between invasion sports with a similar internal logic can be studied through declarative and procedural knowledge. Students can transfer knowledge between different sports by providing common solutions to the same tactical problem. In this sense, the SCAs are more favorable for the transfer of knowledge [16,59]. This research missed studying whether the students who participated in both intervention programmes were able to transfer the acquired knowledge to other invasion sports with the same internal logic. The study of knowledge transfer between common sports is recommended for future research. In addition, it is necessary to study the influence of gender and experiences in learning other sports.

## 5. Conclusions

The results show that the TGAS and DIS intervention programmes induced improvement in the levels of declarative and procedural knowledge. However, there were no significant differences between the groups according to the teaching method used. The heterogeneous nature of the groups regarding gender and experience affected the levels of declarative and procedural knowledge after the application of the intervention programmes. The boys had greater experience and a higher level of knowledge. The students who had no knowledge of football improved equally independently of the method and, those who did have knowledge, improved even more with the TGAS intervention programme. Similarly, the level of declarative knowledge shown by both groups was higher than that of procedural knowledge, which shows knowledge is more centred on the technical aspects than on the behaviours and/or strategies that should be used (tactical decisions). Thus, further research is needed to study the methodologies for the teaching of invasion sports such as football at the primary education stage and their relation with students’ cognitive structures.

## Figures and Tables

**Figure 1 ijerph-17-01063-f001:**
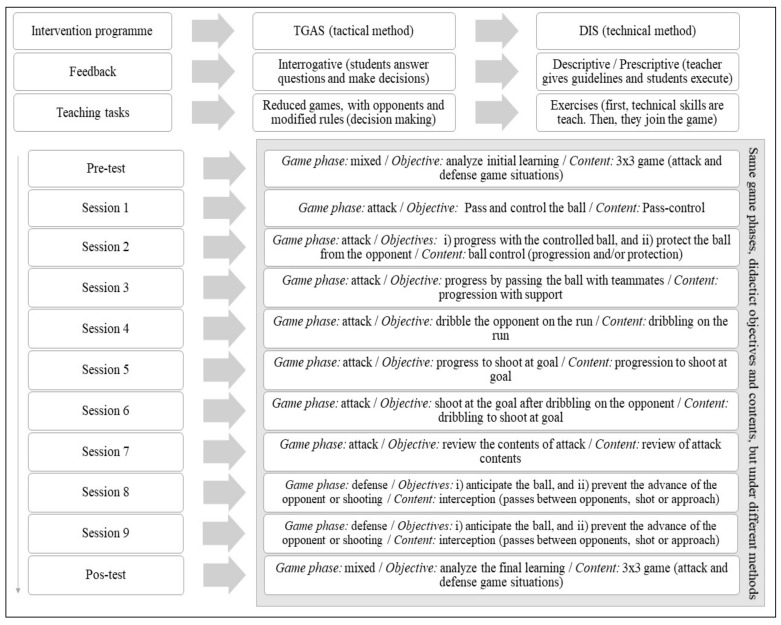
Structure of the TGAS and DIS intervention programmes. Note: TGAS: Tactical Games Approach Soccer. DIS: Direct Instruction Soccer.

**Figure 2 ijerph-17-01063-f002:**
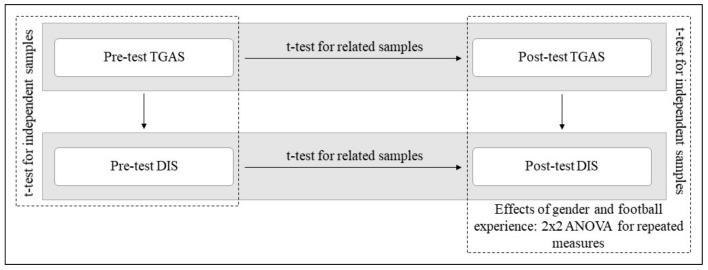
Statistical tests used to compare the declarative and procedural knowledge in the pre-test and post-test. Note: TGAS: Tactical Games Approach Soccer. DIS: Direct Instruction Soccer.

**Table 1 ijerph-17-01063-t001:** Characteristics of the students participating in the study.

Methodology and Class Group	Gender				Experience (Football Practice)			
	Boy		Girl		Yes		No	
	*n*	*%*	*n*	*%*	*n*	*%*	*n*	*%*
TGAS (5 A)	8	40.00	12	60.00	3	15.00	17	85.00
DIS (5 B)	15	71.43	6	28.57	9	42.90	12	57.10

Note: TGAS: Tactical Games Approach Soccer. DIS: Direct Instruction Soccer.

**Table 2 ijerph-17-01063-t002:** Indicators evaluated by the TCTOF.

**Declarative Knowledge**
Individual technical-tactical elements in football, related to action principles for attack. Individual technical-tactical elements in football, not related to action principles for attack. Group technical-tactical elements in football. Action principles for attack in invasion games and sports.
**Procedural Knowledge**
Individual technical-tactical elements in situations of maintaining possession of the ball in football. Individual technical-tactical elements in situations of advancing towards the opponent’s goal in football. Individual technical-tactical elements in situations of scoring a goal in football.

Note: TCTOF: Tactical Knowledge Assessment test in football.

**Table 3 ijerph-17-01063-t003:** Descriptive results of the pre-test and post-test according to gender and football experience.

**Programme**	**Knowledge**	**Gender**	**Pre-test M ± SD**	**Post-test M ± SD**	**Pre-Post**
TGAS (5 A)	Declarative	Boy	45.83 ± 9.69	52.31 ± 16.01	−6.48
		Girl	36.11 ± 9.75	41.05 ± 14.08	−4.94
	Procedural	Boy	34.17 ± 16.11	44.17±15.91	−10.00
		Girl	30.00 ± 14.35	28.33 ± 8.59	1.67
	**Knowledge**	**Experience**	**Pre-test M ± SD**	**Post-test M ± SD**	**Pre-Post**
	Declarative	Yes	49.38 ± 2.14	65.43 ± 5.66	−16.05
		No	38.34 ± 10.71	42.05 ± 13.98	−3.71
	Procedural	Yes	48.89 ± 16.78	60.00 ± 6.67	−11.11
		No	28.63 ± 12.64	30.20 ± 9.46	−1.57
**Programme**	**Knowledge**	**Gender**	**Pre-test M ± SD**	**Post-test M ± SD**	**Pre-Post**
DIS (5 B)	Declarative	Boy	51.11 ± 9.10	57.04 ± 14.47	−5.93
		Girl	37.04 ± 17.37	43.21 ± 14.57	−6.17
	Procedural	Boy	39.55 ± 10.83	48.89 ± 21.48	−9.34
		Girl	22.22 ± 8.07	24.44 ± 9.11	−2.22
	**Knowledge**	**Experience**	**Pre-test M ± SD**	**Post-test M ± SD**	**Pre-Post**
	Declarative	Yes	50.62 ± 8.69	57.20 ± 16.57	−6.58
		No	44.44 ± 15.71	50.00 ± 14.60	−5.56
	Procedural	Yes	42.96 ± 11.60	56.30 ± 23.12	−13.34
		No	28.33 ± 9.90	31.11 ± 13.13	−2.78

Note: M: Mean. SD: Standard Deviation. TGAS: Tactical Games Approach Soccer. DIS: Direct Instruction Soccer.

**Table 4 ijerph-17-01063-t004:** Level of declarative and procedural knowledge between both class groups in the pre-test and post-test.

**Test**	**Knowledge**	**Programme**	***n***	**M ± SD**	***t***	***df***	***p***	***d_cohen_***
Pre-test	Declarative	DIS	21	47.09 ± 13.26	1.882	39	0.067	0.588
		TGAS	20	40.00 ± 10.65				
	Procedural	DIS	21	34.60 ± 12.76	0.681	39	0.500	0.213
		TGAS	20	31.67 ± 14.81				
**Test**	**Knowledge**	**Programme**	***n***	***M*** ***± SD***	***t***	***df***	***p***	***d_cohen_***
Post-test	Declarative	DIS	21	53.09 ± 15.51	1.553	39	0.129	0.485
		TGAS	20	45.55 ± 15.53				
	Procedural	DIS	21	41.90 ± 21.72	1.258	39	0.216	0.393
		TGAS	20	34.67 ± 14.12				

Note: M: Mean. SD: Standard Deviation. *df*: degrees of freedom. TGAS: Tactical Games Approach Soccer. DIS: Direct Instruction Soccer. * *p* < 0.05.

**Table 5 ijerph-17-01063-t005:** Level of declarative and procedural knowledge acquired in each class group (pre-test/post-test).

Programme	Knowledge	*n*	M ± SD	*t*	*df*	*p*	*d_cohen_*
DIS	Declarative pre-test	21	47.09 ± 13.26	−2.274	20	0.034 *	0.411
	Declarative post-test	21	53.09 ± 15.51				
	Procedural pre-test	21	34.60 ± 12.76	−2.368	20	0.028 *	0.339
	Procedural post-test	21	41.90 ± 21.72				
TGAS	Declarative pre-test	20	40.00 ± 10.65	−1.899	19	0.073	0.401
	Declarative post-test	20	45.56 ± 15.53				
	Procedural pre-test	20	31.67 ± 14.81	−1.013	19	0.324	0.207
	Procedural post-test	20	34.67 ± 14.12				

Note: M: Mean. SD: Standard Deviation. *df*: degrees of freedom. TGAS: Tactical Games Approach Soccer. DIS: Direct Instruction Soccer. * *p* < 0.05.

**Table 6 ijerph-17-01063-t006:** Effect of the methodologies on gender and experience in football.

	Knowledge	Variable	*df*	*Quadratic M*	*F*	*p*	*η2*
Gender	Declarative	Intersection	1	32236.844	131.589	0.000 *	0.776
		Gender	1	2671.214	10.904	0.002 *	0.223
		Methodology	1	225.811	0.922	0.343	0.024
	Procedural	Intersection	1	26126.118	77.448	0.000 *	0.671
		Gender	1	4163.195	12.341	0.001 *	0.245
		Methodology	1	2.010	0.006	0.939	0.006
Experience	Declarative	Intersection	1	20135.160	73.660	0.000 *	0.660
		Experience	1	1593.121	5.828	0.021 *	0.133
		Methodology	1	371.978	1.361	0.251	0.035
	Procedural	Intersection	1	26091.556	101.178	0.000 *	0.727
		Experience	1	7182.556	27.852	0.000 *	0.423
		Methodology	1	16.116	0.062	0.804	0.002

Note: *M*: Mean. *df*: degrees of freedom. * *p* < 0.05.

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
