# Peer review of "Experience as a Determinant of Declarative and Procedural Knowledge in School Football"

_ijerph, 2020, doi:10.3390/ijerph17031063_

Round 1
Reviewer 1 Report
The study represents an important contribution to the sparse literature on the topic involved. It presents data from an interesting study. The article is well structured and easy to read. The Abstract section is correct and follows the relevant parts: Purpose, Method, Results, Discussion, Conclusion. I believe that the results are sufficiently relevant, although there are a number of issues (mainly related to the methodological section) that would need to be addressed before publication:
In terms of writing, the paper relies on some references that from my viewpoint could be exchanged with some more recent ones instead. It would be necessary to deepen the following sentence (it appears twice in the paper), as well as to strengthen the text with bibliographic references: “The groups were not modified to maintain the ecological validity of the study.” On the other hand, I suggest to the authors to deepen much more (it would be convenient to offer the reader more details and evidence of the process) on the validation process of the tool mentioned in the following sentence: "Three independent variables were determined: i) (...) and validated by a panel 148 of 13 judges experts (Aiken's V ≥ 0.69; α = 0.97) " One of the most important requirements would be to explain more or justify the sampling, with all the limitations that it presents. It seems one of the weakest points of the study. At least, it would be interesting if this question were explained in the letter of response to the reviewers. In relation to the previous point, it would be interesting to add a section that tells about the main limitations of the study, and of what issues could be improved if someone wanted to replicate the study. Finally, it would be interesting for a native professional to review the language.Overall, it is an interesting study that represents an important contribution to the literature of this field, and it clearly contributes new knowledge to gain insight on how experience determines declarative and procedural knowledge in school football.
Author Response
All manuscript
- First, a native translator performed a grammatical revision of the manuscript (certificate is attached).
- All corrections were marked in red.
- Some references were updated.
Methods
- The process of design and validation of intervention programmes, TGAS and DIS, was deepened (Lines 148-159).
- It is indicated that the Spanish education system does not segregate students according to gender (Line 141). Therefore, the groups were not modified to maintain the ecological validity of the study.
Discussion
- The following sentence was deleted: The groups were not modified to maintain the ecological validity of the study. This sentence is found in the methods section (Line 141-142).
- Some limitations of the study were noted (Lines 377-380).
- New problems were also indicated that can be improved in future studies (Lines 381-389).

Reviewer 2 Report
Possibility of including the variable 'Pre-post difference' in the tables
3 and 4.
The objective of the Primary curriculum is not to teach sports modalities. Therefore, conclusions should not be drawn solely for football ... You should seek transfers with other sports ... See (Memmert and Harvey, 2010; Yañez and Castejón 2011 ...)
It seems important not to direct the study only to the sport modality; rather towards tactical intelligence, decision making etc ...
Why doesn't the student-centered teaching approach get significant differences? .
It is not well explained
Author Response
All manuscript
- First, a native translator performed a grammatical revision of the manuscript (certificate is attached).
- All corrections were marked in red.
Results
- The variable “Pre-Post Difference” was inserted in Table 3. Table 4 compares declarative and procedural knowledge according to the intervention programme applied. Thus, this variable was not inserted.
Discussion
- The effect of students’ gender and football experience on the results was indicated. In this research, the number of boys who participated in the TGAS programme was lower than in the DIS programme. The number of students, all boys, who practised football in the out-of-school context was also lower in the TGAS programme. Therefore, this program did not show significant differences in declarative and procedural knowledge between pre-test and post-test. A sentence was inserted to clarify the previous text (Lines 303-313).
- This research focused on comparing teaching methodologies, Tactical Games Approach and Direct Instruction, by teaching football in physical education. However, the transfer of knowledge acquired to other invasion sports with the same internal logic was not analyzed. It was indicated as a problem to study in future research (Lines 381-389).

Reviewer 3 Report
- Language review by a native.
Clarify the conclusions based on the objetives. some limitations about the study review the study sample To consult: EJSS.
Author Response
All manuscript
- First, a native translator performed a grammatical revision of the manuscript (certificate is attached).
- All corrections were marked in red.
Discussion
- Some limitations of the study were noted (Lines 377-380).
- New problems were also indicated that can be improved in future studies (Lines 381-389).
Conclusion
- It was indicated that, after implementing the intervention programmes, there are no differences depending on the teaching methodology due to the influence of gender and the football experience of students. It was also indicated that students showed higher levels of declarative than procedural knowledge.

Round 2
Reviewer 1 Report
.